# Understanding Challenging Behaviors in Autism Spectrum Disorder: A Multi-Component, Interdisciplinary Model

**DOI:** 10.3390/jpm12071127

**Published:** 2022-07-12

**Authors:** Stephen M. Edelson

**Affiliations:** Autism Research Institute, San Diego, CA 92116, USA; director@autism.com; Tel.: +1-619-281-7165

**Keywords:** medical comorbidities, interoception, challenging behaviors, self-injurious behavior, self-harming behavior, aggression, autism spectrum disorders

## Abstract

A multi-component, interdisciplinary model is described which explains the presence of, and in other cases the lack of, many challenging behaviors associated with autism spectrum disorder (ASD). More specifically, the model expands the operant behavioral conditioning paradigm by taking into account medical comorbidities and interoceptive processing.

## 1. Introduction

Challenging behaviors commonly associated with autism spectrum disorder (ASD) include aggression toward others, self-injurious (or self-harming) behaviors, and severe tantrumming [1,2]. Many individuals with ASD exhibit multiple challenging behaviors; for example, a survey of 2327 individuals on the autism spectrum found that more than 40% engaged in both aggression and self-injurious behavior (SIB) [3].

Aggression may include scratching, biting, hitting, or kicking [4,5]. SIB may include excessive scratching or rubbing, hair-pulling, hand-biting, headbanging, or face-slapping. Severe tantrumming may sometimes include one or more of these behaviors. All of these behaviors vary in frequency, duration, and severity across the autism spectrum [6].

Treatment of these behaviors has been only moderately successful [7]. Laverty and her colleagues conducted a follow-up survey of 67 ASD individuals who engaged in SIB on a regular basis and found that 44% still engaged in these behaviors 10 years later [8].

Over the past 50 years, researchers have identified operant conditioning, and more recently medical comorbidities, as major contributors to challenging behaviors [9,10,11]. In this model, medical and behavioral factors as well as impaired interoception are utilized to explain why many ASD individuals engage in challenging behaviors. In addition, implications for assessment and treatment are discussed.

## 2. Operant Conditioning

The operant (or instrumental) conditioning paradigm of challenging behaviors has received a great deal of experimental support beginning in the 1960s [11,12,13,14]. Basically, this model states that an antecedent (or stimulus) provokes a behavior. Soon after, the behavior is reinforced, positively or negatively, by one or more consequences. As a result, the behavior will more likely occur under similar circumstances in the future. The traditional paradigm is:

Antecedent > Behavior > Consequence(s)

Researchers have investigated specific types of antecedents and consequences that trigger and later maintain behaviors, and have found that these typically can be attributed to the actions of other people in the vicinity. For instance, individuals may want to escape a situation (e.g., a demand) or obtain something (e.g., attention from others or access to a preferred tangible item) [11,12,15,16,17].

Carr and colleagues studied a psychotic child in both demand (e.g., “Point to the window”) and non-demand (e.g., “The birds are singing”) situations and observed a dramatic increase in SIB following demands but only a slight increase after non-demands [15]. In a related study, Edelson et al. [16] observed 20 ASD individuals, ages 6 to 20 years, over a five-hour period, and recorded all antecedents and consequences of a wide range of SIB. A total of 19 of the 20 individuals engaged in SIB after a staff intervention (i.e., demand, denial, verbal punishment), whereas one child exhibited chin banging that had no temporal relationship to any form of social interaction.

Social attention given contingent on challenging behaviors may reinforce them and lead to an increase. In a study conducted by Lovaas and Simmons [18], sympathetic comments, such as “I don’t think you are bad”, were given contingent on a challenging behavior; this resulted in an increase in the frequency and magnitude of the behavior.

Based on this behavioral perspective, researchers and clinicians began to treat challenging behaviors by changing the consequences. Approaches included punishment techniques (e.g., blindfold/facial screening [19], electric shock [20], and inhalation of ammonia [21]) and ignoring the behavior (i.e., “extinction”) [13].

The operant conditioning paradigm also takes into account the relationship between an underlying biological or medical condition and a challenging behavior. For example, an ear infection can be an antecedent to headbanging [22], and repetitive stereotyped behaviors may, in some cases, be intrinsically biologically rewarding (i.e., consequence) [23,24,25].

## 3. Motivational Factors and Setting Events

Based on the writings of Bijou and Baer [26], Carr and colleagues expanded the basic operant conditioning paradigm by taking into account the individual’s internal condition and external surroundings [11,27]. These are referred to as “setting events.” Examples of internal or biological setting events include constipation [28], fatigue [29], menstrual pain [30], and otitis media [31]. Examples of environmental/physical setting events include lighting, sound, and temperature. The setting event in relation to the operant paradigm is:

Setting event > Antecedent > Behavior > Consequence

For example, a student who typically finds it aversive to work on math exercises in a classroom may occasionally engage in aggression to escape working on such assignments. However, if the child is suffering from a medical comorbidity, such as a stomachache, then working on the math assignment will become even more aversive. As a result, the student may be more inclined to escape the task. In this example, the stomach pain is the setting event. In other words, the increased aversiveness of the demanding task makes escaping the math assignment more reinforcing and increases the likelihood of aggressive behavior.



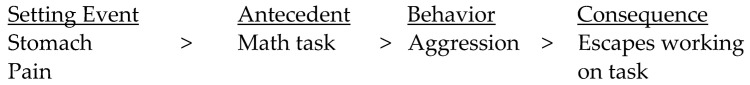



Numerous medical comorbidities have been associated with autism, including allergies, anxiety, constipation, gastroesophageal reflux disease (GERD), migraines, otitis media, rhinitis, sinusitis, and sleep disturbances [10,32,33,34,35]. Research has also documented that many of these comorbidities co-occur with various challenging behaviors [3,9,36,37,38]. 

In one study, Smith et al. [29] documented an increase in challenging behaviors in relation to fatigue (setting event) *and* demands (social antecedent) in three ASD individuals. In another study, four females with intellectual disabilities were given demands during times with and without menstrual pain (setting event). The results clearly showed an increase in behavioral problems, such as aggression, SIB, and tantrumming, during times of menstrual pain [30].

In this model, a challenging behavior may continue even after a medical comorbidity has been resolved. For example, Carr et al. reported on a 10-year-old boy who scratched himself because of a skin allergy. After the skin healed, his scratching behavior continued and was maintained by social attention [17].

Also, a medical comorbidity may be either an antecedent or a setting event. Short-duration medical conditions, such as the sudden onset of a stomachache or headache, may trigger a behavior (i.e., antecedent). In contrast, a relatively long-lasting condition, such as poor-quality sleep [39], may be considered a setting event.

## 4. Interoception

Interoception refers to the perception of internal sensations within the body, including bladder fullness, bowel movements, discomfort/pain, heartbeat, hunger, respiration, and thirst [40]. The anterior insula and the ventromedial prefrontal cortex are responsible for interoceptive processing, and several studies have reported impairments in these neural structures in autism [41,42]. Consistent with these findings, published controlled experiments [43,44] and self-reports [45,46] have documented dysregulated interoceptive processing in many individuals on the autism spectrum.

Garfinkel et al. [43] found that many of these individuals displayed an “exaggerated” or a *hyper*-response to internal bodily sensations. Schauder et al. [47] also documented that ASD individuals attend to internal sensations for longer periods of time compared to neurotypical controls. Thus, intense awareness of internal distress may increase the likelihood that discomfort or pain becomes a setting event or an antecedent to a challenging behavior.

Research has also indicated a tendency for some individuals on the autism spectrum to be *hypo*-responsive with respect to interoceptive processing [45,46]. These individuals are sometimes described as having a high threshold for discomfort or pain. As a result, they may be less aware, or even unaware, of an internal illness or condition; thus, they may be less likely to experience discomfort or pain.

A third form of interoceptive impairment involves perceiving that something is wrong internally but being unable to locate the area of discomfort or pain [48]. The unpleasant feelings caused by this awareness of internal distress, combined with the inability to identify its source, may be a setting event.

Examples of how this model can explain the presence (and in other cases, the absence) of a challenging behavior are illustrated below.



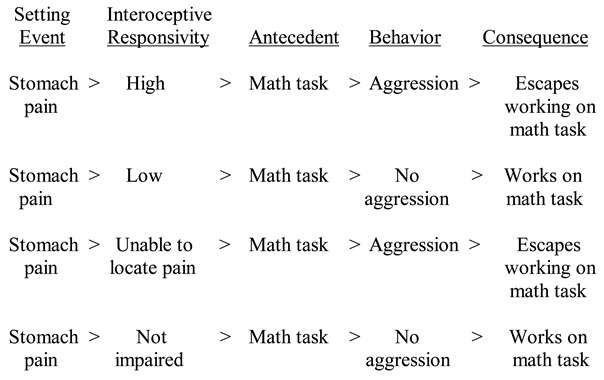



It is important to mention that some challenging behaviors may be a direct result of an underlying biological impairment. Examples include the relationships between low serotonin levels and aggression [49], seizures and SIB [50], and calcium deficiency and eye-poking [51]. The proposed model would not likely apply in these circumstances.

## 5. Interoception and Anxiety

Anxiety is one of the most prevalent medical conditions associated with autism, and is estimated to be a significant problem for 40% to 80% of the ASD population [52]. Anxiety has also been associated with challenging behaviors, such as aggression, SIB, and tantrumming [53].

For more than a decade, researchers have studied anxiety and interoception [54]. Anxiety may be an antecedent to a challenging behavior in cases involving a sudden onset; for instance, an individual may exhibit severe tantrumming as a result of social anxiety stemming from an unanticipated social interaction. In other circumstances, anxiety may be a setting event in which the individual suffers from a long-lasting, sometimes chronic, form of anxiety. Such anxiety may be a result of a dysregulated autonomic nervous system [55], medical and/or nutritional issues [56,57], or possibly exposure to environmental toxins [58]. In the latter case, Edelson et al. proposed that certain toxins, such as particulate matter, pesticides, and heavy metals, may trigger an immune response (cytokine activity), which in turn dysregulates the autonomic nervous system and leads to anxiety. These factors may also be mediated by the form and degree of interoceptive impairment.

## 6. Assessing Individuals with Challenging Behaviors

Proper assessment using valid and reliable evaluation tools is critical to understanding challenging behaviors and developing an appropriate treatment strategy to reduce or eliminate them. Much research has supported the efficacy of conducting a functional behavioral assessment to document physical and social antecedents and consequences of the behavior in question [59]. Clearly, a thorough medical assessment is equally important, given the possibility of short-term or chronic illnesses involving a multitude of bodily functions such as body temperature, heart rate, and metabolism [60].

Although medical comorbidities are well-documented in the autism research literature, a comprehensive assessment protocol has yet to be formalized. Regarding gastrointestinal distress, clinicians and researchers often rely on parent checklists [61,62] in addition to laboratory blood work, stool studies, and endoscopy [63,64]. Immune conditions, such as sinusitis and rhinitis, are often assessed by measuring antibody levels, such as Immunoglobulin E (IgE)-mediated allergic disorders [65].

Sleep disturbances can be examined by employing parent checklists [66] as well as extensive polysomnography assessments such as brain wave activity, heart rate and rhythm, and oxygen saturation [67]. Anxiety assessments have relied on observing behaviors associated with anxiety such as fidgeting, hand-wringing, and pacing [14,68,69] as well as administering parent checklists [70], and in some cases, measuring levels of autonomic nervous system activity [71].

Admittedly, diagnosing pain or illness in individuals with ASD can often be difficult, given the communication challenges of these individuals as well as the interoceptive impairments common among this population. However, the correct diagnosis and treatment of medical problems that contribute to challenging behaviors is vital, because this can markedly improve the quality of life for individuals with ASD as well as their caregivers, and may result in less restrictive academic, vocational, and residential placements.

Assessment tools often used to evaluate the form and degree of interoceptive impairment have been developed for the neurotypical population, but their validity and reliability have been mixed [72,73,74] Although there are recommended ways to identify problems with interoception in ASD individuals, valid and reliable tools have not yet been published. However, Mahler [48] (pp. 38, 40, 42) has suggested a number of observable behaviors that may likely indicate the type of impaired interoceptive processing. Examples of hyper-responsiveness include “requests bathroom breaks very frequently” and “seems always to be hungry and/or thirsty.” Examples of hypo- or under-responsiveness include “never feels hungry and/or thirsty” and “may have a fairly significant health issue and never complains of symptoms (e.g., strep throat, urinary tract infection, broken finger, fever).” Examples of difficulty pinpointing the location of distress include “complains of feeling sick but cannot provide any specific symptoms” and, when asked if a bathroom is needed, “replies, ‘I don’t know’ or ‘maybe.’”

Regarding setting events, external variables, such as air temperature and noise level, are relatively simple to record. However, biological setting events, such as anxiety, chronic constipation, and fatigue are more difficult to observe. Thus, extensive questioning of individuals with ASD (when possible) and their caregivers is critical.

In addition, clinicians and caregivers should schedule frequent medical check-ups and monitor the health of individuals with ASD on a regular basis. In particular, clinicians need to be aware that, as mentioned earlier, discomfort or pain may not be perceived by those with hypo-responsive interoception. These individuals should be monitored especially closely, since some medical comorbidities may require urgent care (e.g., bone fractures, chronic constipation, dental pain, ear infections, GERD).

Whenever possible, individuals with ASD should be taught to communicate areas of discomfort or pain (e.g., saying “it hurts” or pointing to sensitive areas). Also, care providers should be taught how to identify vocalizations (e.g., gagging moaning, whining/sobbing) and/or behaviors (e.g., flinching, holding/rubbing a body part, repeatedly pulling the ear, wincing) associated with discomfort or pain.

It may also be helpful to teach individuals with ASD to better recognize interoceptive sensations [75,76]. For example, Schaefer et al. [77] tested a heartbeat perception training procedure on non-autistic outpatients and documented a reduction in anxiety.

For their part, researchers should examine the relationships between various medical conditions and physiological correlates, such as arousal level, body temperature, heart rate, and respiration. Understanding the complexity of challenging behaviors is vital to the health and welfare for many of these individuals.

## 7. Conclusions

It is often difficult to identify the underlying contributors to challenging behaviors. As a result, these behaviors frequently persist into adulthood [8,78]. In many instances, individuals who exhibit these behaviors may be experiencing discomfort or pain associated with a medical comorbidity, and the form and extent of their interoceptive impairment may determine their level of suffering. Further investigations on the relationship among antecedents and consequences, medical comorbidities, setting events, and interoception will allow researchers to develop a comprehensive, evidence-based battery of assessments in order to assist clinicians and therapists in developing effective treatment strategies to treat challenging behaviors.

## Data Availability

No applicable.

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
