# Peer review of "Understanding Challenging Behaviors in Autism Spectrum Disorder: A Multi-Component, Interdisciplinary Model"

_jpm, 2022, doi:10.3390/jpm12071127_

Round 1
Reviewer 1 Report
In this interesting work is described a multi-component, interdisciplinary model which explains challenging behaviors associated with autism spectrum disorder. Taking into account the crucial importance of the correct diagnosis and treatment of medical problems that contribute to challenging behaviors, I suggest the Author mention a clinical evaluation protocol and blood/urinary/fecal examinations protocol, if any, to better and earlier identify medical problems.
Author Response
I added two paragraphs (with 12 references) on evaluating several medical comorbidities. See page 6, lines 196-207.
Reviewer 2 Report
As a review, this article provides valuable information but there was no novel findings or conclusion.
The integrity of the manuscript should be improved. Also, the reviewer suggests that some sentences in the introduction should be cited for example Line 17 ends with tantrumming, need to be referenced. Also, line 20,22
Author Response
Correct, there were no novel findings reported in this paper. I hope this paper will provide researchers with further insight into the underlying reason(s) for challenging behaviors. I also described various assessments that could be employed in future studies (pages 6 and 7, lines 188-238).
Regarding the conclusion, I rewrote the final sentence in the manuscript in which I stated the goal of studying possible contributors to challenging behaviors: "Further investigations on the relationship among antecedents and consequences, medical comorbidities, setting events, and interoception will allow researchers to develop a comprehensive, evidence-based battery of assessments in order to assist clinicians and therapists in developing effective treatment strategies to treat challenging behaviors." (page 8, lines 265-269).
More references were added (page 1, lines 17, 20,22).
Regarding the integrity of the manuscript, I would be happy to improve the integrity of the paper. However, I would appreciate some suggestions.